# The Crossroads between RAS and RHO Signaling Pathways in Cellular Transformation, Motility and Contraction

**DOI:** 10.3390/genes12060819

**Published:** 2021-05-27

**Authors:** Olga Soriano, Marta Alcón-Pérez, Miguel Vicente-Manzanares, Esther Castellano

**Affiliations:** 1Tumor Biophysics Laboratory, Centro de Investigación del Cáncer and Instituto de Biología Molecular y Celular del Cáncer, Consejo Superior de Investigaciones Científicas (CSIC)-University of Salamanca, 37007 Salamanca, Spain; olgaasoriano@gmail.com; 2Tumour-Stroma Signalling Laboratory, Centro de Investigación del Cáncer and Instituto de Biología Molecular y Celular del Cáncer, Consejo Superior de Investigaciones Científicas (CSIC)-University of Salamanca, 37007 Salamanca, Spain; maralcper2@usal.es

**Keywords:** Ras, Rho, myosin II

## Abstract

Ras and Rho proteins are GTP-regulated molecular switches that control multiple signaling pathways in eukaryotic cells. Ras was among the first identified oncogenes, and it appears mutated in many forms of human cancer. It mainly promotes proliferation and survival through the MAPK pathway and the PI3K/AKT pathways, respectively. However, the myriad proteins close to the plasma membrane that activate or inhibit Ras make it a major regulator of many apparently unrelated pathways. On the other hand, Rho is weakly oncogenic by itself, but it critically regulates microfilament dynamics; that is, actin polymerization, disassembly and contraction. Polymerization is driven mainly by the Arp2/3 complex and formins, whereas contraction depends on myosin mini-filament assembly and activity. These two pathways intersect at numerous points: from Ras-dependent triggering of Rho activators, some of which act through PI3K, to mechanical feedback driven by actomyosin action. Here, we describe the main points of connection between the Ras and Rho pathways as they coordinately drive oncogenic transformation. We emphasize the biochemical crosstalk that drives actomyosin contraction driven by Ras in a Rho-dependent manner. We also describe possible routes of mechanical feedback through which myosin II activation may control Ras/Rho activation.

## 1. Introduction

Small GTPases are molecular switches that cycle between active GTP-bound and inactive GDP-bound forms and regulate a variety of cellular signaling events, including growth, cellular differentiation, cell motility and survival. Arguably, the best characterized of the small GTPases belong to the Ras superfamily and include five main family members: Ras, Rho, Ran, Rab, and Arf. These families can be further divided into subfamilies that share the common core G domain providing essential GTPase and nucleotide exchange activity to regulate discrete cellular processes [1,2]. Members of the Ras subfamily regulate cell proliferation and differentiation, whereas members of the Rho subfamily control the actin cytoskeleton, gene expression and proliferation.

Aberrant Ras signaling is found in up to 30% of all human cancers [3]. Even 40 years after the initial discovery of Ras oncogenes in 1982, no approved drug directly targets Ras in Ras-driven cancer. New information and approaches for direct targeting of mutant Ras fueled hope for the development of direct K-Ras inhibitors [4] and have begun to challenge the perception that Ras is undruggable. The development of direct Ras inhibitors that specifically target the glycine-to-cysteine mutation at residue 12 (G12C) in K-Ras has been a breakthrough in the field. As a result, different RasG12C-specific inhibitors have entered phase 1/2 clinical trials in solid tumors presenting K-RasG12C mutation: AMG 510 (NCT03600883 and NCT04185883), MRTX-849 (NCT03785249), JNJ-74699157 (NCT03114319), and LY3499446 (NCT04165031), with more in the pipeline. Early clinical trial observations have been promising, particularly for Non Small Cell Lung Carcinoma (NSCLC) [5,6].

Ras proteins are small GTPases bound to the plasma membrane that cycle between the GDP- and GTP-bound states as a consequence of the stimulation of certain cell surface receptors, such as the epithelial growth factor receptor (EGFR). The conversion from inactive GDP-bound forms to the active GTP-bound form is stimulated by guanine nucleotide-exchange factors (GEFs). Conversion back to the inactive form is mediated by GTPase-activating proteins (GAPs) [7]. GEFs and GAPs are large, multi-domain proteins capable of an astonishing variety of interactions with other proteins, lipids, and regulatory molecules that control the cellular levels of active and inactive Ras [8].

In the GTP-bound form, Ras switches into a conformation where it can bind and activate effector proteins, building different signaling networks at the membrane of cells, bridging extracellular cues into cellular events that regulate almost any aspect of cellular physiology, such as cell growth, proliferation, differentiation, and survival, having a decisive role in the acquisition of an oncogenic phenotype by healthy cells [9]. Activating mutations in Ras proteins diminishes their ability to hydrolyze the bound GTP to GDP, thus rendering it constitutively active even in the absence of receptor engagement.

The list of well characterized Ras effectors is long (Figure 1), but the Raf/MAPK and PI3K/AKT cascades are the two best characterized downstream effectors of Ras signaling, due to their implication in many Ras-driven forms of cancer [10,11]. 

Rho GTPases are highly homologous to Ras [12]. They also act as GTP-dependent molecular switches that regulate cell dynamics, cell growth and tissue development. RhoA, Rac1 and Cdc42 are the most-studied members of the Rho family. Of these, only RhoA and Rac1 appear significantly mutated in a narrow spectrum of human cancers [13], although alterations to their levels of expression are frequent. Despite having many functions, RhoA is best known for its ability to promote the formation of actin stress fibers and focal adhesions, whereas Cdc42 and Rac regulate the formation of filopodia and lamellipodia, respectively. They all are heavily involved in cell migration through the control of actin dynamics [14,15,16]; in addition, they modulate gene expression, cell cycle progression, and cell survival [17,18]. RhoA, B and C and Rac1 play critical roles in cell transformation induced by activated, oncogenic Ras. Dominant negative Rho and Rac1 constructs inhibit Ras-induced transformation. Conversely, constitutively active variants promote anchorage-independent growth and other features of the transformed phenotype [19,20,21,22,23]. Pioneering work revealed that the requirement of Rho for Ras-induced transformation is due, at least in part, to the fact that Ras and Rho play opposing roles in control of the cyclin-dependent kinase inhibitor p21WAF1/CIP1. Indeed, Ras promotes p21WAF1/CIP1 transcription, whereas Rho inhibits it. Increased Ras activity blocks cell cycle progression when Rho signaling is inhibited by C3 exoenzyme [22,24,25]. Similarly, Rac activation is upregulated in H-Ras-transformed rodent fibroblasts [26,27,28]. These results were the basis of other studies, in which dominant-negative Rac1 mutants, which sequester and inactivate Rac-GEFs, impaired the growth of H-Ras-transformed rodent fibroblasts [23,29,30]. Experiments using genetically modified mouse models revealed that tissue-restricted genetic loss of Rac1 impaired mutant K-Ras-driven lung [31] and pancreatic [32] cancer development. Furthermore, in a mutant K-Ras-driven model of papilloma development, tumor tissue exhibited increased levels of Rac–GTP, and loss of one Rac1 copy alone was sufficient to reduce tumor growth and increase survival [33]. In melanocytes, Rac1 with Rac1 P29S cooperates with oncogenic B-Raf and Nf1 to promote melanoma development and resistance to therapy [34]. 

Cell movement is essential to many biological phenomena, including embryogenesis, wound healing, and cancer metastasis. It is a multistep process in which cells extend membrane protrusions in the cell front (lamellipodia, filopodia and/or membrane blebs), interact with the extracellular matrix or neighboring cells, contract the cell body and detach the cell rear to enable net movement of the whole cell. Migration depends on the tight regulation of the actin cytoskeleton reorganization by Rho GTPases. The dynamics of actin in migrating cells are based on the different architecture of actin-based subcellular structures, which depends on the differential segregation and/or activation of specific adaptors, e.g., the Arp2/3 complex, or myosin II. The details of these processes have been extensively reviewed elsewhere [35,36,37,38]. 

Cancer cells use different modes of migration and can transition from one mode to another during cancer progression and metastasis. In lamellipodium-based migration, lamellipodium and filopodium formation involves new actin polymerization, and requires actin nucleators such as the Arp2/3 complex, activated by Rac and Cdc42. Actin polymerization in lamellipodia can also be mediated by formins after RhoA activation [39]. Formation of integrin-based focal adhesions by Rho/ROCK signaling stabilizes lamellipodia and mediates interaction with the extracellular matrix. Actomyosin-based contractility and detachment of the cell rear are controlled by Rho/ROCK signaling. In bleb-based migration, actomyosin contractility driven by RhoA and/or RhoC activation of ROCK leads to the extension of blebs, which are protrusions of the cell membrane caused by actomyosin contraction of the cell cortex. ROCK, together with MLCK, acts by increasing the phosphorylation of myosin light chain, which increases the interaction of myosin II with actin filaments [40,41].

The aberrant activation of Ras proteins has been implicated in facilitating virtually all aspects of the malignant phenotype of the cancer cell, including cellular proliferation, transformation, invasion and metastasis. It is now clear that oncogenic Ras signaling also regulates the functions of other Ras-related proteins to promote tumor progression, such as cell invasion and metastasis formation [42]. Given the multitude of cellular activities on which tumor metastasis relies, it is not surprising that oncogenic Ras promotes this aspect of the transformed phenotype by engaging several effector networks. Ras-dependent effector pathways that have been shown to collaborate with RAS in metastasis progression include the Ras–MAPK, Ras–PI3K, Ras–Ral GTPase and Ras–Rho GTPase pathways [43,44]. Each of these pathways can promote tumor progression and metastasis at multiple steps. For example, the activation of Rho GTPases leads to concurrent alterations in cell adhesion and cell motility [45] and loss of Ras proteins severely restricts cell motility and migration in fibroblasts and cause major alterations in cytoskeletal structures [46]. Furthermore, in some settings, the induction of metastasis is the product of cooperation between oncogenic Ras and other metastasis-involved pathways, such as the TGF-β pathway [47,48]. Using Caco-2 cells as a model, Makrodouli et al. demonstrated that mutations in H-, K-Ras or B-Raf induced different changes in cell morphology due to regulation of different Rho GTPases [49]. They demonstrated that mutations in BrafV600E promoted a significant increase in cell migration and invasion through activation of RhoA GTPase in a MERK–ERK-dependent fashion. However, K-Ras V12D mutants enhanced the ability of Caco-2 cells to migrate and invade through filopodia formation and PI3K-dependent Cdc42 activation. Finally, H-Ras V12D mutants increased cell migration and invasion via Rac1 activation, along with the mesenchymal morphology obtained through the EMT. Moreover, B-Raf and K-Ras oncogenes were shown to cooperate with the TGF-β1 pathway to provide cells with additional transforming properties. Defining the precise Ras/effector-dependent mechanisms that control the metastatic capacity of tumor cells requires a better understanding of the context-dependent outcome of their coordinated activation.

In this review, we will focus on the connections between Ras and Rho signaling as they converge on the protein myosin II, a major generator of mechanical force in normal and cancer cells. 

## 2. Ras–Rho Crosstalk through PI3K Activation

Phosphoinositide 3-kinases (PI3Ks), Ras and Rho GTPases are key regulators of cell polarization, motility and chemotaxis. They influence each other’s activities by direct and indirect feedback processes that are only partially understood (Figure 2). 

Ras can bind to and activate class I PI3Ks through the Ras Binding Domain (RBD) present in all p110 catalytic subunits, except p110β, despite containing an RBD domain [50]. Initial studies showed that in *Drosophila*, mutation of the Dp110 RBD impaired activation with Ras 1 (homolog of mammalian RAS isoforms) and severely reduced insulin responses, body size and egg numbers [51]. Experiments using neutrophils revealed that defective p110 RBD significantly reduced activation of PI3K and migration of neutrophils [52]. A great deal of our current understanding in the regulation of PI3K activation by Ras proteins come from the RBD models generated by Downward and colleagues. In these models, Ras binding to p110α is abolished by introduction of two point mutations in the RBD domain of the catalytic subunit [53]. Using this model, they reported that in primary mouse embryonic fibroblasts (MEFs), AKT phosphorylation was significantly impaired in response to epithelial growth factor (EGF) and completely abrogated in response to fibroblast growth factor (FGF2). They also reported that immortalized MEFs lacking Ras binding to PI3K present impaired motility in response to EGF, hepatic growth factor (HGF), insulin and oncogenic Ras signaling, which could be rescued by introduction of an active AKT construct in the cells [54]. Regarding tumor development, the RBD model demonstrated that Ras binding to PI3K is essential for the initiation of lung and skin tumors and abrogation of this interaction halted initiation of both tumor types [48] and greatly impaired tumor maintenance [55]. 

Once PI3K is activated by Ras, it triggers Rac activation, and actin rearrangement correlates with the ability of Ras mutants to activate PI3K. Inhibition of PI3K activity blocks Ras induction of membrane ruffling, while activated PI3K is sufficient to induce membrane ruffling, acting through Rac [56]. Blockade of Ras binding to PI3K decreases fibroblast motility induced by EGF (but not platelet-derived growth factor (PDGF)) as it curbs Rac activation [54]. Additionally, EGF-dependent phosphorylation of AKT phosphorylates Rac1 in S71, promoting the interaction of 14-3-3 proteins with Rac1, which seems to regulate Rac1 localization [57]. However, the functional relevance of this interaction is not fully understood yet. Activated PI3K enhances actin polymerization and membrane protrusion during cell motility via activation of Rac1, Cdc42 and RhoG [56,58].

Once activated, class I PI3Ks produce PtdIns(3,4,5)P3, which functions as a second messenger that recruits proteins containing phosphoinositide-binding modules, such as the PH domain. PH domains have been identified in several members of the GEF family, including the Dbl family of Rho-GEFs [59,60]. The identification of a PH domain capable to bind membrane phosphoinositides in different Rho-GEFs provided a possible mechanism whereby cytosolic proteins, such as GEFs, are translocated to the plasma membrane upon receptor activation [61]. For example, Tiam1 presents two different PH domains, one at the N terminus and the other one at the C terminus. The N-terminal PH region interacts with PtdIns(3,4,5)P3 and enhances Rac1 activation in vivo and in vitro [62]. Rac activation drives formation of protrusions at the leading edge of migrating cells that are formed by the localized polymerization of actin: active Rac (GTP-bound) interacts with and activates the Wave regulatory complex (WRC). Active WRC triggers the Arp2/3 complex to support branching of new actin filaments [63].

PI3K cooperates with Rac to regulate cell polarization in response to different stimuli to define the leading edge of the migrating cells [64]. While the initial reports on the interaction between PI3Ks and Rac were obtained using neutrophils and *Dictyostelium* cells, this mechanism is conserved in other cell types with different migratory modes [65,66]. In unstimulated cells, PtdIns(3,4,5)P3 are almost absent, but increases rapidly and dramatically after stimulation with chemoattractants [67] and couples receptor binding with Rac activation to promote actin polymerization. Consistent with this, pharmacological inhibition of PI3K activity, or dominant-negative PI3K proteins, results in defective actin polymerization and cell polarity [68,69,70]. Phosphatase and TENsin homolog (PTEN) and SH2 domain containing inositol-5-phosphatase (SHIP1) antagonize PI3K-dependent activation of Rac signaling, further contributing to the establishment of an asymmetric distribution of PtdIns(3,4,5)P3 in migrating cells [71,72], although by different mechanisms. PTEN and SHIP1 dephosphorylate the 3’ and the 5’ phosphate, respectively, of PtdIns(3,4,5)P3, leading to the formation of PtdIns(4,5)P2 or PtdIns(3,4)P2 [73,74]. The localization of PTEN and PI3K are mutually exclusive, facilitating the accumulation of PtdIns(3,4,5)P3 at the trailing edge, SHIP1 is active at the cell-matrix interface to abolish the gradient of PtdIns(3,4,5)P3 that are induced by integrin activation [75]. PTEN is a binding partner of the Rac-specific GEF P-REX1.

The role of different Ras family members in the activation of the different PI3K isoforms and its connection to Rho GTPases is not fully understood. Yang et al. reported that, although several Ras family members could activate PI3K activity, only H-, K-, N- and R-Ras (R-Ras, R-Ras2 and R-Ras3) directly binds to the RBD domain of p110 and act as robust PI3K activators [56]. In contrast, several Rho family small GTPases activated PI3K by an indirect cooperative positive feedback that required a combination of Rac, Cdc42 and RhoG small GTPase activities. Instead, p110β is a direct target for Rac and Cdc42. Rac1 is essential for p110β activation downstream of the GPCRs for LPA and S1P, with the Rac-GEF Dock180/Elmo1 being upstream of both Rac1 and p110β in this pathway, which is highly conserved. In this way, p110β controls Rac-dependent actin remodeling, migration and phagocytosis [76]. Thus, a complex network of Ras and Rho family small GTPases induces and reinforces PI3K activity, explaining the relevance of different small GTPases in regulating PI3K to control cell polarization and chemotaxis.

While most published work suggests that Rac acts downstream of PI3K [54,56,77,78,79,80], several studies indicate that PI3K can be also activated by Rho GTPases [50,56], thus creating a positive feedback loop that would potentiate PI3K activation, accumulating active Rac at the cell front [45,81,82]. Actin polymerization mediated by small GTPases contributes to this positive feedback loop, likely by recruiting adaptors to the right subcellular localizations [79,83,84,85,86]. The combined activation of Rac, Cdc42 and RhoG may be critical for positive feedback on PI3K activation [56]. Activation of individual small GTPases or knockdown of a single downstream activator failed to trigger PI3K activation, despite inducing actin polymerization. In contrast, PI3K was activated upon the simultaneous activation of endogenous Rac, Cdc42 and RhoG by Vav2. The degree of PI3K activation was dependent on the number of downstream activators, although the amount of actin polymerization by these GEFs was similar. The concept of cooperative activation of PI3K, by a combination of multiple upstream Ras family member inputs and positive feedback amplification by several Rho family small GTPases, is generally accepted [87,88,89]. After external stimulation, GEFs for upstream activators are recruited to the plasma membrane and switch the upstream activators to an active state. The upstream activators then bind to and recruit PI3K to the plasma membrane for the production of PtdIns(3,4,5)P3, which, in turn, recruit PH-domain-containing Rho-GEFs, including Tiam1 and Vav2, to the plasma membrane [56,77,78]. This results in the activation of downstream GTPases, which induce actin polymerization and generate a positive feedback loop that further activates PI3K in a cooperative manner. 

Deleted in liver cancer 1 (DLC1) is a Rho-GAP protein that negatively regulates Rho GTPases and is associated to focal adhesions [90]. After RTK stimulation (for example by EGF, IGF-1 or insulin), the kinase activity of AKT is necessary to elevate the levels of GTP-bound RhoA via DLC1 both in transformed and non-transformed cell lines that express DLC1, but not in those with downregulated DLC1. In DLC1-positive cells, AKT phosphorylated three serines in DLC1 (S298, S329 and S567), which greatly attenuated its Rho-GAP and tumor-suppressor activities [91]. The phosphorylation of the serine residues by AKT induced strong binding between the linker region and the Rho-GAP domain, which places DLC1 in a closed, inactive conformation that is mainly monomeric and reduced RhoA-GTP, tensin and talin binding to DLC1 and its subsequent localization to focal adhesions [91].

Another point of interconnection between Ras and Rho involves the hetero-oligomerization of Rho-GDP and Ras-GTP [92]. Using *Dictyostelium* as model, Senoo et al. reported that chemoattractant-induced phosphorylation of GDP-Rho promotes its hetero-oligomerization with GTP-Ras. The Rho–Ras hetero-oligomers directly activate mTORC2 toward AKT phosphorylation in cells and in vitro. In contrast, Rho-GTP inhibits the mTORC2-AKT signaling by blocking Rho–Ras hetero-oligomerization. Interestingly, they found that K-Ras 4B functionally replaces *Dictyostelium* Ras in the activation of mTORC. In this way, mTORC2-AKT may regulate actin cytoskeleton and cell-substrate adhesion to control directed cell migration toward chemoattractants [37,93,94,95].

## 3. Ras–Rho Crosstalk through Raf/MEK/ERK Activation

Although less studied than PI3K, indirect signaling from Ras to Rho GTPases through the Raf/MEK/ERK pathways seems to be a common occurrence as well. Formation of focal adhesions and reorganization of the actin cytoskeleton in pancreatic [96] and ovarian [97] cancer cells are dependent on active Ras/Raf/MEK/ERK signaling. ERK directly regulates the actin polymerization machinery to stabilize protrusions into fast, prolonged events to overcome membrane tension and promote motility [98,99]. Increased protrusion is accompanied by modulation of adhesion and contraction dynamics for productive motility. Indeed, ERK promotes adhesion disassembly [100,101] and myosin activation [102,103,104], but the mechanisms and molecular roles of this regulatory circuit are not fully elucidated yet. Interestingly, Barbacid group found that in MEFs lacking Ras proteins (Rasless), only constitutive activation of the components of the Raf/Mek/ERK pathway was sufficient to sustain normal migration. In a Rasless context, activation of the phosphatidylinositol 3-kinase (PI3K)/PTEN/AKT and Ral guanine exchange factor (RalGEF)/Ral pathways, either alone or in combination, failed to induce migration, although they cooperated with Raf/Mek/ERK signaling to reproduce the full response mediated by Ras signaling [46].

Upon growth factor stimulation, active ERK co-localizes with the Wiskott–Aldrich Syndrome Protein-family verprolin homologous protein 2 (WAVE2) at the cell edge at the time of active protrusion, promoting the interaction of WAVE with Arp2/3 [98]. ERK activation promoted protrusion reinforcement by Arp2/3-mediated actin polymerization but was not needed for cell protrusion initiation. ERK localized to the cell edge phosphorylates and activates the WAVE regulatory complex for further Arp2/3 recruitment and activation [99]. This effect generates the actin polymerization power needed to overcome increasing membrane tension and to augment protrusion velocity and persistence.

Enrichment of ERK at focal adhesions in both spreading and adherent cells is mediated by its association to the scaffolding protein receptor for activated C kinase 1 (RACK1) [105,106,107]. RACK1 promotes adhesion activation of ERK that, in turn, induces local suppression of p190-RhoGAP from the cell periphery [108]. ERK localization to large and stable adhesion plaques induces sustained ERK signaling and p190 depletion, leading to an increase in RhoA activity, which in turn further stabilizes remaining adhesions. Permanent depletion of p190 from the cell periphery results in the formation of actin bundles, reduced protrusion and cell rear formation during migration [109].

Recent work in a melanoma cell model suggested that, in addition to p190-RhoGAP regulation, ERK controls focal adhesion remodeling by curbing Rnd3 expression [110]. Targeted knockdown of B-Raf with small interfering RNA or pharmacological inhibition of MEK increased actin stress fiber formation and stabilized focal adhesion dynamics due to stimulation of the Rho/ROCK/LIM kinase-2 signaling pathway, cumulating in the inactivation of the actin depolymerizing/severing protein cofilin. Constitutive expression of Rnd3 suppressed the actin cytoskeletal and focal adhesion effects mediated by B-Raf knockdown and depletion of Rnd3 elevated cofilin phosphorylation and stress fiber formation and reduced cell invasion, suggesting that Rnd3 is a key mediator of this crosstalk.

Raf-1 has been heavily scrutinized as a downstream effector linking Ras activation to the MEK/ERK module. A role for Raf-1 in the regulation of the cytoskeleton during migration has also been established, involving binding to the Rho effector ROCK2 in a kinase-independent manner. Work from Manuela Baccarini’s group has shown that binding of Raf-1 to ROCK2 reduces the kinase activity of ROCK2 and is essential for the proper regulation of adhesion and migration in several cell types [111,112,113], including blocking ROCK2-induced keratinocyte differentiation in epidermal tumorigenesis [114,115,116] and modulating ROCK2 activity at vascular endothelial (VE)-cadherin junctions during tumor-induced angiogenesis [113,117]. Binding of Raf-1 to Ras is necessary for the interaction between Raf-1 and ROCK2, and the formation of the Raf-1/ROCK2 complex requires the cysteine-rich domain (CRD) of Raf-1 [112]. They have proposed an elegant model to explain this mechanism, which posits that recycling part of the Raf-1 molecules that are activated in the context of ERK to act on ROCK2 creates diversification in the signaling network, linking mitogenic Ras signaling with cytoskeletal rearrangements [118]. The model also proposes that different Raf-1 species are formed depending on differential residue phosphorylation. Raf-1 species that interacts with ROCK2 lack phosphorylation on Ser621, which depends mostly on Raf-1 autophosphorylation [119]. After RAS activation, the B-Raf/Raf-1 dimer is formed and Raf-1 autophosphorylates on Ser621, in a positive feedback loop that keeps itself in the ERK pathway; ERK promotes dimer dissociation, reducing the kinase activity of RAF-1. Two phospho-species may be produced: if dephosphorylation of Ser621 prevails, then pSer338 and pSerTri (Ser289/Ser296/Ser301), containing species that associate with ROCK2, is produced; in contrast, if pSer338 dephosphorylation prevails, the desensitized Raf-1 form that contains pSerTri and pSer621 is produced and dissociates from the signaling complex to return to the cytosol. It is still unclear how this decision is made, but this model potentially defines the timeline of ERK activation during cell proliferation, and the interaction of Raf-1 with ROCK2 to promote cytoskeletal rearrangements.

The balance between ERK–MAPK and Rho-kinase signaling is a key determinant of vascular morphogenesis [120]. ERK–MAPK signaling promoted endothelial cell survival and vessel sprouting by downregulating Rho-kinase activity. In macrophages, CCL2-induced activation of MAPK during migration is accompanied by activation of RhoA GTPase activity and actin polymerization [121]. Studies in colon cancer [122,123] and melanoma [110] cells demonstrated that constitutive K-Ras signaling inhibits Rho activation and actin stress fiber formation in a MEK-dependent manner, although the mechanism is different; whereas, in colon cancer cells, MEK-regulated expression of Fra-1 reduced actin stress fiber formation by inactivating β1 integrins, and in melanoma cells, B-Raf regulates stress fiber formation and focal adhesion turnover by control of Rnd3, which inhibits the Rho/ROCK/LIM kinase/cofilin signaling pathway.

Another level of ERK regulation over Rho GTPase function is through control of Rho-GEFs and GAPs. Activation of ERK promotes phosphorylation of the Rho-GAP DLC1 in Ser129, facilitating the binding of SRC to DLC1 and promoting further phosphorylation of DLC in Tyr451 and Tyr701. Thus, SRC and ERK1/2 signaling converge on DLC1 to cooperatively regulate it, attenuating the Rho-GAP and tensin-binding activities of DLC1 [124]. A more complex regulation of Rho-GEF and GEF-H1 (ARHGEF2) by ERK has been suggested, in which, by phosphorylation of different residues of GEF-H1, ERK controls RhoA activity. ERK phosphorylate GEF-H1 on Thr678, enhancing its nucleotide-exchange activity toward RhoA [125] downstream of stress signaling, such as TNF-α and membrane depolarization [126]. It is possible that upon activation of stress signals, several pathways are activated, which results in the formation of an ERK/Scaffold/GEF-H1 complex that enables ERK to phosphorylate Thr678 efficiently. However, under growing conditions, ERK has been shown to phosphorylate GEF-H1 in Ser959, reducing its GEF activity and decreasing RhoA function. In MDA-MB-231 cells, MEK inhibition or mutations on the Ser959 residue of GEF-H1 have a dramatic effect on cells in 3D matrices: cells become rounded and resemble cells with an amoeboid migration pattern. This shift in morphology and to a lesser extent the deficient motility can be rescued by either suppressing RhoA signaling or reducing endogenous levels of GEF-H1, suggesting that the transition from mesenchymal to amoeboid morphology can be induced by GEF-H1. In this way, GEF-H1 may easily be inactivated by high ERK activity when mesenchymal invasion is preferred. On the other hand, high levels of GEF-H1 activity will facilitate transition to an amoeboid type of invasion [127]. Thus, by modulating phosphorylation of GEF-H1, ERK would enable to maintain high invasive motility regardless of its environment.

Direct phosphorylation may constitute yet another mechanism by which ERK activation can modulate Rac1 and RhoA activity. ERK can phosphorylate Rac1 in Thr108 in response to EGF stimulation and this phosphorylation decreases Rac1 activity, targeting Rac1 to the nucleus and altering its role in mediating cell migration [128]. This phosphorylation is mediated by direct interaction of the ERK docking site in the Rac C-terminus. ERK has also been shown to phosphorylate RhoA at Ser88 and Thr100 by direct interaction at the D-site on the C-terminus of RhoA. This phosphorylation enhances RhoA activity and its function in mediating stress fibers formation by favoring its interaction with ROCK1 (without affecting its interaction with mDIA) [129]. By phosphorylating both RhoA and Rac1, ERK is able to increase RhoA activity and decrease Rac1 activity. This mutual antagonism produces balanced activities of RhoA-generated apical constriction and Rac1-dependent cell elongation to control cell shape, and thus, the curvature of the invaginating epithelium.

## 4. Ras–Rho Crosstalk through GEFs and GAPs

GAPs and GEFs are the main regulators of Ras and Rho signaling. Studies based on GEFs and GAPs established multiple points of crosstalk between both signaling pathways. By interacting with different specific partners, GAPs and GEFs appear to endow cells with functional plasticity to respond to evolving conditions, both intra- and extracellular [130]. Many GEFs and GAPs contribute to Rho GTPase-mediated migration. However, the dynamic regulation of Rho GTPases needed for cells to migrate in response to changes in their environment requires the coordinated and localized activation/inactivation of multiple components, rather than a simple linear interaction between GEFs/Rho-GTPases/effectors/GAPs [18].

### 4.1. GEFs

The activity of Rho GTPases is spatially regulated in many cellular functions, following the subcellular localization of GEFs. PI3K activation plays an essential role in the regulation of GEF localization through the production of PtdIns(3,4,5)P3, which binds to the PH domain of GEFs [18]. Rac-GEFs are well-known effectors of EGFR and other RTKs and, in many instances, trigger the activation of different Rho-GEFs in a PI3K manner [131,132]. Considering the reliance of K-Ras on Rac1 in cancer development, different Rac-GEFs have been considered as K-Ras effectors, potentially via a PI3K-dependent mechanism [131]. Here, we will focus on Sos and Tiam1, the two best-characterized GEFs connecting Ras and Rho GTPases.

Son of Sevenless (Sos) is a dual specificity GEF that regulates both Ras and Rho family GTPases and thus is uniquely poised to integrate signals that affect both gene expression and cytoskeletal reorganization. Two isoforms, Sos1 and Sos2, have been found, presenting similar protein structures and cellular expression patterns, but specific functionalities (recently reviewed by Baltanas et al. [133]). Although the molecular mechanisms of Ras-GEF catalysis have been uncovered, how Sos1 acquires Rac-GEF activity and what is the physio-pathological relevance of this activity is less understood. Structural and biochemical studies revealed that Sos1 is a multi-domain protein with extensive intramolecular interaction that tightly constrains its activity [134]. The differential activity of Sos1 over Ras or Rac targets appears to be mostly mediated by mutually exclusive interactions with either Grb2 or E3B1 adaptor proteins. When the GEF-Sos1 is in complex with Grb2, it activates Ras, whereas when complexed with Eps8, it activates Rac1 [125,130]. Activation of Rac1 by Sos1 involves the formation of SH3-mediated Sos1-E3B1 complexes that are recruited to actin filaments found within membrane ruffles in an Eps8-dependent manner, facilitating the Sos1-GEF activity over Rac1 [135,136]. Once active, Rac1 triggers activation of downstream effectors, enabling actin cytoskeleton remodeling [137]. Disruption of the Eps8–E3B1–Sos1 complex by genetic removal of Eps8 or by dominant-negative E3B1 abrogate Rac1 activation and Rac1-dependent actin remodeling induced by RTKs, Ras or PI3K [138,139]. Formation of the Eps8–E3B1–Sos1 complex has been shown to be PI3K-dependent, since it requires interaction of p85 (the regulatory subunit of PI3K) with E3B1 [140]. Interestingly, p66SHC specifically activates Rac1 by reducing formation of the Sos1–Grb2 complex (activating Ras) and increasing the formation of the Sos–Eps8–E3B1 complex that specifically targets Rac1 [141]. It was reported that phosphorylation of Sos1 at the Y1196 in the C-terminal is sufficient to cause its Rac-GEF activity in response to the activation of various receptors and TKs and contribute to BCR–ABL-induced leukemogenesis [142].

Activated Ras interacts with the Ras-binding domain of the Tiam1 protein at the plasma membrane [26,143]. Tiam1 is a Rac exchange factor directly activated by Ras-GTP through a Ras-binding motif in its N terminus. Tiam1 is mainly involved in the regulation of Rac1-mediated signaling pathways, including cytoskeletal activities, cell polarity, endocytosis and membrane trafficking, cell migration, adhesion and invasion, cell growth and survival, metastasis and carcinogenesis [144]. Initial studies of Tiam1-dependent activation of Rac in cancer showed that it was required for Ras-induced skin tumors due to an increased apoptosis rate in epidermal keratinocytes during tumor initiation [26,145]. Tiam1 is predominantly cytoplasmic, but in response to cellular activation it can be translocated to the plasma membrane. Membrane translocation has been reported to be critical for Tiam1-induced membrane ruffles and activation of Rac1 effectors [144]. Tiam1 can associate with PI3K products although its membrane localization seems to be independent of this association. Rac activation via Tiam1 leads to binding and phosphorylation of PAK serine threonine kinases (p21-activated kinases) [146]. Such interaction promotes changes in actin cytoskeletal dynamics, including the polymerization of actin filaments at the leading edge of the cell to form lamellipodia and ruffles. PAK exerts its effects on many key regulatory proteins via phosphorylation, e.g., PI3K, Raf and β-catenin, and it modulates cell growth and survival; however, it acts predominantly on cytoskeleton rearrangement and cell migration [147]. At adherens junctions, phosphorylated Tiam1 displays a docking site for the Grb2–Sos complex, which increases activation of ERK, a kinase that is constitutively associated with Tiam1. ERK activation then triggers localized degradation of Tiam1, leading to a reduction in Rac activity and weakening of cell–cell adhesion, allowing increased cell migration [148]. Our current knowledge indicates that Tiam1 regulates different aspects of tumor development, promoting tumor progression or antagonizing tumor invasion, making therapeutic intervention challenging [144].

### 4.2. GAPs

As stated above, Rho-GAPs turn Rho GTPases off by stimulating their intrinsic rate of GTP hydrolysis. Rho-GAPs are multi-domain proteins with various protein and lipid interactive domains capable of precise targeting and regulation in signaling complexes. Some have more than one catalytic activity and potentially signal to other pathways in addition to terminating Rho signals. In this respect, Rho GTPases provide spatial and temporal information to GAPs that in turn serve as protein adaptors in a variety of intracellular compartments [18].

Rottapel group work on the Rho-GEF ARHGEF2 has uncovered an important link between Ras and Rho in cancer. The Rho GEF ARHGEF2 (also known as GEF-H1) is a microtubule-associated guanine exchange factor and Dbl family member demonstrating exchange activity toward RhoA [149]. ARHGEF2 contributes to cell survival and growth in K-Ras- and H-Ras-transformed cells both in vivo and in vitro [150,151] by establishing a feedback loop that amplifies the MAPK pathway. ARHGEF2 was shown to mediate Ras transformation in *Drosophila* as well, promoting tissue overgrowth and invasion [152]. *Arhgef2* is transcriptionally activated downstream of K-Ras and the *Arhgef2* promoter is transactivated downstream of multiple signaling pathways, including MAPK and PI3K [153]. Mitogenic signals conveyed through the MAPK pathway might be coupled to microtubule function via ARHGEF2, thereby coordinating growth signals with changes in cell shape, migration and/or morphogenesis. ARHGEF2 was also shown to function as a mediator of the K-Ras oncogene addiction and the cell motility gene signature present in a panel of PDAC cell lines [154]. Thus, ARHGEF2 could be a therapeutic target to increase the efficacy of MAPK pathway inhibitors for treatment of pancreatic cancer. 

Certain GAPs may serve as effector proteins of Ras signaling, coupling Ras to Rho GTPases. They utilize a catalytic arginine finger to stimulate the inefficient intrinsic GTP hydrolysis reaction of these small FTP-binding proteins by several orders of magnitude [155]. p120RasGAP (p120), p190ARhoGAP (ARHGAP35), p200RhoGAP (ARHGAP32) and DLC1 (or ARHGAP7) act as linkers between the Ras and Rho signaling pathways [80,156,157]. p120 contains multiple domains with different functions [158]. Whereas the C terminus of p120 with the catalytic GAP activity is responsible for Ras inactivation [159,160], its N-terminal SH2 and SH3 domains possess an effector function [161,162,163,164,165]. p120 functionally modulates Rho signaling by binding to two Rho-specific GAPs, p190 and DLC1 [80,166,167]. The association of p120 with the tyrosine phosphorylated p190 via its SH2 domain promotes Rho inactivation, positively regulating the Rho-GAP function of p190 [167,168,169,170]. p120’s SH3 domain binds DLC1 by interacting with its Rho-GAP region, inhibiting its GAP activity by targeting its catalytic arginine finger. Functional characterization and structural elucidation of the trans-inhibitory mechanism of DLC1 mediated by p120 protein indicated that the GAP activity of DLC was almost completely abolished in the presence of the SH3 domain. Overexpression of DLC isoforms lead to inactivation of RhoA and to the reduction of actin stress fiber formation, suggesting that DLC proteins are Rho-selective GAPs and the role of the DLC trans-inhibitory protein p120 is to retain Rho proteins in their active GTP-bound states [165]. 

p120 interacts with a third Rho-GAP, p200RhoGAP. In contrast to p190 and DLC1, which are downstream of p120, p200RhoGAP has been proposed to bind to the p120 SH3 domain via its very C-terminal proline-rich region and to sequestrate its Ras-GAP function, preventing it from inactivating Ras [157]. Besides its C-terminus Ras activation effect, p200RhoGAP also uses its N-terminal Rho-GAP domain to downregulate endogenous RhoA activity, thereby facilitating the GDP/GTP cycling of RhoA to promote cell proliferation. 

Nf1 is another Ras-GAP that links the Ras and Rho signaling pathways. Unlike p120, Nf1 does not interact with a Rho-GAP, instead interacting with two downstream effectors of RhoA, LIMK1 and LIMK2. By interacting with the SecPH domain of Nf1, LIMK2 is not activated by its upstream effector ROCK and partially loses its kinase activity on cofilin [171,172]. By interacting with the pre-GRD region, Nf1 negatively regulates the Rac1/Pak1/LIMK1/cofilin pathway [173]. Cofilin promotes actin depolymerization and severs the long actin filaments, which leads to a fast turnover of actin filaments [174,175]. In this manner, the Ras-GAP Nf1 may coordinate the inhibition of both LIM kinases shutting down their activity on cofilin, thus promoting actin stress fiber formation. The regulation of these two branches of the pathway by two independent domains suggests an independent regulation for these processes. Independently of the mechanism, by inhibiting cofilin, Nf1 plays a central role in the regulation of actin cytoskeleton reorganization, contributing to diverse cellular functions such as cell motility, morphogenesis, division, differentiation, apoptosis, neurite extension and oncogenesis.

In addition to cofilin regulation, other specific signaling pathways were identified to connect Nf1 to actin cytoskeleton dynamics, especially cell adhesion. Nf1 overexpression induces an increase in the expression of the FAK [176] and direct interaction between Nf1 and FAK has also been described [177]. Moreover, by interacting with syndecan-2, another adhesion protein, Nf1 mediates the activation of PKA, which phosphorylates two actin-associated proteins (Ena and VASP), thus promoting actin polymerization for the formation of filopodia and dendritic spines [178].

## 5. Ras–Rho Crosstalk through Ral

Ral GTPases (RalA and RalB) coordinate membrane trafficking and actin polymerization. Additionally, Ral proteins are central transducers of the oncogenic Ras signaling. Some direct activators of Ral (RalGDS, Rgl1, Rgl2 and Rgl3) are Ras effectors, supporting the existence of a Ral–Ras axis. Activated GTP-bound Ras recruits these Ral-GEFs at the plasma membrane, triggering the activation of Ral by GDP to GTP exchange. Indeed, the Ral pathway is a key player in human cancer progression, particularly in invasion and metastasis. Of the two mammalian Ral orthologs, RalA, is associated with tumor initiation and anchorage-independent growth, whereas RalB is associated with invasion, metastasis and survival [156,179,180,181]. Initial indications that Ral proteins had a role in cell migration were obtained in *Drosophila*, where expression of a dominant negative Ral delayed the onset of migration of a group of specific follicle cells that migrate in the egg chamber during the fly oogenesis (border cells) without affecting migratory speed [182]. Subsequent studies showed that Ral activity was locally detected at lamellipodia, suggesting a crucial function for Ral in promoting protrusion formation [183]. Definitive studies showing the relevance of Ral proteins in migration were provided using gene silencing approaches, showing that RalB, but not RalA, was required for the migration of both normal [184] and cancer cells [185], pointing out non-overlapping and possibly even antagonistic functions of the two Ral proteins as they regulate cell motility.

Ral activates three main oncogenic effectors. One of these, RalBP1, is implicated in the generation of invadopodia [186]. The other two, Exo84 and Sec5, are components of the hetero-octameric exocyst complex. The exocyst works in polarized exocytosis and is at the center of multiple protein–protein interactions that support cell migration by promoting protrusion formation, front-rear polarization and extracellular matrix degradation [187]. Indeed, the exocyst shuttles between trafficking intracellular vesicles and the plasma membrane and, furthermore, it physically interacts with many regulators of the cytoskeleton, including the main components of the Rho family, Rac, RhoA and Cdc42 [179]. The exocyst is also required for motility as cell movement requires membrane delivery to the actively expanding plasma membrane at the leading edge. RalB has been implicated in migration of normal rat kidney cells [184], bladder cancer cells [185] and B-lymphocytes [188]. Fluorescence resonance energy transfer analysis revealed that RalA is highly activated at lamellipodia in migrating MDCK cells [183]. In addition to membrane lipid components, Ral and the exocyst deliver α5-integrin to the focal adhesion at the leading edge through interaction with paxillin [189], which enables adhesion of the migrating cells to extracellular matrix proteins, such as fibronectin.

Several molecular links connect the exocyst to the Rac–WRC–Arp2/3 pathway during migration: (i) different subunits of the exocyst complex (EXOC4/Sec8 and EXOC8/Exo84) bind to the Rac-GAP SH3BP1, inactivating Rac1-GTP at the front to promote the continuous oscillation between the GDP-bound and GTP-bound conformations [190,191]; (ii) EXOC7/Exo70 and EXOC3/Sec6 bind to the WRC complex, contributing to the trafficking of WRC toward the edge of nascent protrusions [192]; and (iii) EXOC7/Exo70 binds to and stimulates activity of the Arp2/3 complex, promoting actin branching [193]. EXOC7/Exo70 also interacts with PI(4,5)P2 phospholipid and induces plasma membrane curvatures, contributing to protrusion formation and migration. Once at the leading edge, WRC may be activated by the basal levels of Rac-GTP and membrane PtdIns(3,4,5)P3. Then, the exocyst complex dissociates from WRC and is recycled in the cytosol. The positive feedback loop between actin cytoskeleton and Rac [194,195] would lead to sustained Rac activity and stronger WRC stimulation. Finally, EXOC2/Sec5 interact with paxillin, targeting the exocyst to focal complexes where it might contribute to the turnover of cell adhesions to the substrate [189].

Ral has also been implicated in invasion and metastasis. It was reported that 3T3 cells transformed by the RasV12G37 mutant (a Ras effector mutant that activates Ral-GEFs but not Raf or PI3K [196]) formed aggressive lung metastasis following tail-vein injection. The metastatic activity of RasV12G37-transformed cells was inhibited by expression of a dominant negative RalB [197]. RalB, but not RalA, is required for the contractility-driven invasion of lung cancer cells (A549, K-Ras mutated) [198]. Moreover, in vivo metastasis assays in mice (tail vein injection) [197,199] and in hamsters (subcutaneous injection) [200] supported a function of RalB pathway in the formation of tumor metastasis, both in Ras-mutated and Rous sarcoma virus-transformed cells. These studies demonstrated the relevance of the RalB pathway for motility, invasion and metastasis, but the underlying molecular mechanisms remain elusive. Recently, it has been proposed that active Ras binds and recruits the two Ral-GEFs RGL1 and RGL2, which activate RalB. Activated RalB binds to the exocyst complex, promoting its assembly and recruitment to the leading edge; by its direct association with the WRC, exocyst drives WRC to the leading edge, where WRC stimulates actin polymerization, protrusion formation, motility and invasion [201].

## 6. Ras, Rac and Mechanosensing

Organs and tissues are in constant motion, exposing epithelial cells to mechanical stretch. However, how these external forces impact cellular morphology, organization and dynamics in healthy and diseased tissues is still being elucidated. Increase in the rigidity of the microenvironment in which oncogene-expressing cells are embedded may be sufficient to ignite oncogenic mechano-signaling and empower oncogene-mediated cell reprogramming.

Panciera and colleagues have recently shown that oncogenic K-Ras (and RTKs) “sense” the rigidity of the ECM to rewire cell-intrinsic mechanical parameters through activation of the YAP/TAZ transcriptional programme, hence bestowing tumorigenic potential to normal cells harboring oncogenic mutations [202]. Their work points to Rac1 as a crucial mediator of cellular mechanical parameters downstream of K-Ras and HER2 activation in response to stiff ECM, implicated in breast and pancreatic cancers [203]. The gain of function of Rac1 closely mimicked the pro-oncogenic programming in breast and pancreatic organoids that was abolished by YAP/TAZ depletion, strategically placing Rac1 as an upstream effector for YAP/TAZ-mediated cytoskeletal and oncogenic reprogramming for enhancing the crosstalk between the cells and the rigid environment. 

In the same line, it has recently been reported that in epithelial cells under static condition, the specific configuration of the oncogenic Ras and WT machinery favors apical extrusion (the mechanism by which a dying cell is eliminated, a cellular process that is also part of normal cell competition [204,205]). A perturbation to this system, via mechanical stretching, abolishes the formation of the actin belt around oncogenic Ras cells and potentiate its protrusion. The addition of ROCK inhibitor in a mechanically dynamic microenvironment partially restores the cortical actin belt around RasV12 cells, enabling a degree of apical extrusion and inhibiting protrusion formation. This indicates that cyclic stretching drives the system toward RasV12 invasiveness through the activation of cytoskeleton-modifying proteins, at least in part by activating the Rho pathway [206].

Ras directly controls cell geometry, mechanics and force generation during mitosis [207,208,209]. By using non-transformed cells transfected with an inducible RasV12 expression plasmid, it has been shown that short-term activation of Ras induces differences in cell shape during mitosis in an ERK-dependent manner [207]. Although the specific molecular mechanisms have not yet been fully elucidated, these changes appear to depend on actomyosin contractility. This fits with the established role of the Ras/ERK pathway in controlling interphase actin filament organization and contractility to promote cell motility and invasion [99,122,210], as well as to direct the modulation of the RhoA activity [22,49,109,129] and myosin phosphorylation [103,104]. High levels of K-Ras expression in lung cancer cells induces myosin-dependent cell shape changes in the interphase with cells adopting a rounded, sphere-like morphology [209]. Recent work has shown that activation of Ras in pancreatic epithelial cells leads to changes in the apico-basal distribution of myosin and interphase cell shape in a way that contributes to cancer morphogenesis [208].

## 7. Myosin II as an Endpoint of Chemical and Mechanical Signaling

Cell migration plays a pivotal role in a wide variety of phenomena throughout phylogeny. Coordinated cell movement requires protrusive forces generated by polymerization of actin filaments at the leading edge and contractile forces via myosin motors at the rear of the cell. Contraction forces generated by myosin II motor proteins [211] are coordinated with actin polymerization at the leading edge and have several roles in migration [36,212].

Myosin II (the text refers throughout to non-muscle and smooth muscle myosin II, not skeletal or cardiac) is regulated by a myriad of signals that control its function at different levels, including conformational extension, catalytic activity and filamentation [36]. The most-studied mechanism consists of phosphorylation of the regulatory light chain of myosin II in Ser19, which promotes the conformational extension of the functional unit of myosin II, the hexamer [213]. It also increases the ATPase activity of the head domain [214]. 

Several signaling pathways control Ser19 phosphorylation. Among these, the best characterized stem from the small GTPase RhoA. Active RhoA expression in fibroblasts promoted extensive stress fiber and focal adhesion formation [215] in a myosin II-dependent manner [216]. Among the downstream effectors of RhoA, S/T ROCK and CITK phosphorylate and activate myosin II [217]. In addition, ROCK also phosphorylates MYPT1, which is a myosin II phosphatase [218]. Hence, ROCK is a dual activator of myosin II, phosphorylating myosin II directly and also inactivating MYPT1, which would dephosphorylate myosin II (Figure 3). Other kinases can also phosphorylate myosin II directly, including MLCK (Figure 3). MLCK was the first kinase described to phosphorylate myosin II [219]. Its preferential substrate is smooth muscle myosin II, over non-muscle myosin II [220]. In non-muscle cells, MLCK seems to phosphorylate a small population of myosin II that assembles behind the lamellipodium in protruding cells [221,222]. Another important kinase in this context is myotonic dystrophy-related Cdc42-binding kinase (MRCK), which functions downstream of Cdc42 [223] and controls myosin II assembly at the lamellae [224]. 

Finally, ZIPK/DAPK3 is a death-related kinase that can activate myosin II regulatory light chain (RLC). Although the mechanism is less well-described, death-associated protein kinase 3 (DAPK3) can be phosphorylated by DAPK1 during apoptosis [225] and it also interacts with RhoD, an atypical Rho-GTPase [226]. A recent study also indicated that DAPK3 is activated downstream of the RhoA/ROCK circuit [227]. Myosin II function is also controlled by phosphorylation of the RLC in Ser1, which decreases its affinity for actin and its ATPase activity [228]. Phosphorylation of Tyr155 downstream of the RTKs also impairs the formation of the functional hexamer [229]. Finally, myosin II assembly is also regulated by phosphorylation of the coiled-coil and non-helical tail domain. This has been extensively reviewed elsewhere [36,230].

## 8. Role of RAS–MAPK in Myosin II Activation and Adhesion Dynamics

Oncogenic H-Ras profoundly affects the morphology of different cell types [122,231,232]. In mesenchymal cells, activated H-Ras decreases stress fiber formation [231,232], which is a hallmark of myosin II inactivation [216]. This is likely due to the ability of Ras to activate Rac, as activation of the latter has similar effects [16] and is rooted on the classic Rac/RhoA antagonism described early by Collard et al. [233], and later confirmed in various cellular systems [234].

Surprisingly, Cheresh and co-workers also demonstrated that active ERK phosphorylated MLCK (Figure 3), increasing myosin II phosphorylation and its activation, which is at odds with the observations described previously. This mechanism was essential for cell migration in collagen 3D environments [235] and survival by upregulation of integrin/FAK signaling [236]. These contradictions still linger in the field, making it impossible to make general statements as to whether H/K-Ras are bona fide inhibitors of myosin II activity. However, and given the controversy around the actual contribution of MLCK to myosin II regulation [237], it is possible that the cornerstone of these contradictions lie in the actual functional levels of MLCK in each cell line. In cells with high levels of MLCK, e.g., smooth muscle cells, MLCK phosphorylation by ERK could be a dominant mechanism through which H/K-Ras could mainly activate myosin II. In cells in which the MLCK levels are low, the RhoA/ROCK pathway would be dominant towards myosin II activation and repressed by Rac/RhoA antagonism when H/K-Ras is activated.

In addition, most of these effects are Ras isoform-dependent. Activating mutations of R-Ras have an opposite effect to those in H/K-Ras, increasing focal adhesion maturation through a mechanism that depends on p130CAS, but not the canonical Ras effector Ras [238]. Interestingly, R-Ras localizes itself to focal adhesions, unlike H- or K-Ras [239]. This strongly indicates that functional signal divergence enables R-Ras to activate myosin II, whereas H/K-Ras inhibit it. Early evidence confirmed that the signaling routes activated by the isoforms are different. Microinjection of active R-Ras did not activate the ERK, JNK or p38 pathways, while H/K-Ras did [240]. However, R-Ras did activate αMβ2 integrin in macrophages in a Rap1-dependent manner. In fact, Rap1 plays a fundamental role in integrin activation in different cell lineages. The mechanism, which depends on RIAM and kindlin [241], drives the ligation of talin to the tail of the beta chain of several integrin receptors, connecting the integrin to actin as part of a mechanosensitive molecular clutch that connects adhesion size and dynamics with adhesive force, actin polymerization and organization and cell migration [242]. In this regard, talin exposes vinculin-binding sites when mechanically stretched [243], reinforcing the linkage of integrins with actin in a force-dependent manner. A possible mechanism involves R-Ras activating Rap1 to create integrin-actin linkages, which then can be reinforced by myosin II-dependent force application to talin (Figure 3).

## 9. Connecting PI3K to Cellular Contractility

In amoeboid-crawling cells, type I PI3K and its metabolic product PtdIns(3,4,5)P3 denote the front edge of polarized cells. PI3K’s functional opposite, the phosphatase PTEN, localizes to and defines the rear of the cell [37]. Whereas this spatial segregation is less clear in mesenchymal cells, the same fundamental principle applies. PI3K and PtdIns(3,4,5)P3 localize to the leading edge [244,245,246], whereas PTEN and myosin II are specifically excluded from the leading edge [37,247,248]. This suggests that there is a functionally antagonistic relationship between PI3K and myosin II. Following the same train of thought, PTEN should somehow activate myosin II. However, the activation mechanism does not seem direct. In *Dictyostelium* cells, recruitment of PTEN points from external force application preceded that of myosin II. In PTEN-null cells, myosin II recruitment was decreased, suggesting a mechanosensitive function for PTEN in myosin II translocation to the cortex [249]. Likewise, PTEN is required for myosin II to define the sides and rear of polarized *Dictyostelium* cells [250]. Whether this was dependent on PIP2 was not established. Strikingly, not many studies have addressed this connection in mammalian cells. 

It is difficult to establish a hierarchy between type I PI3K/PTEN and myosin II due to the fact that perturbations to PI3K, PTEN or myosin II have profound effects on the morphology of most cells [37]. Some evidence points to a direct role for PI3K/AKT in myosin II activation, for example in response to 2-arachidonoylglycerol [251] or TRPV4-dependent calcium fluxes [252]. Likewise, PI3K inhibition promotes dilation of human small airways, which strongly suggests that PI3K lies upstream of the RhoA/ROCK/myosin II axis in these cells [253]. A recent study showed that constitutive activation of PI3K and RTK signaling in a glioma cell model in *Drosophila* led to the activation of Drak, a close relative of the myosin II kinase DAPK3/ZIPK [254] that phosphorylated Sqh, the *Drosophila* ortholog of myosin II regulatory light chain [255]. 

Other studies have pointed to alternative mechanisms through which PI3K could influence myosin II activity. A recent study described that PI3Kalpha signaling impairs MYPT1 phosphorylation downstream of NUAK1, actively decreasing actomyosin contractility. While this has been shown only in endothelial cells, its extension to other models could constitute a mechanism through which PI3K actively inhibits myosin II activation at the front [256] (Figure 4). Several studies on the regulation of lamellipodial signaling suggest an interesting paradox. Bear and colleagues have established that, in mesenchymal cells, myosin II inactivation at the front is achieved by PKCα-dependent phosphorylation of RLC in Ser1/2 [257] (Figure 4). However, work from Peter Parker’s group indicated that PKCα curbs the lipid kinase activity of PI3K through the phosphorylation of the catalytic subunit [258] (Figure 4). Where these two reports clash is that, based on the aforementioned data, PI3K in pro-protrusive and hence should participate in myosin II inactivation at the lamellipodium, as suggested above. However, PKCα activation at the leading edge would not only block myosin II via Ser1 phosphorylation as reported, but also PI3K, questioning the role of type I PI3K in mesenchymal chemotaxis.

On the other hand, type II PI3K may play different roles in myosin II activation due to their specific compartmentalization. A very recent study has shown that class II PI3K isoforms lie upstream of the small GTPase RhoA (hence myosin II too, which is downstream of RhoA) during vascular smooth muscle contraction and regulates blood pressure [259]. This effect is likely independent of the barrier function of endothelial cells, which seems independent of PI3K, but dependent on myosin II and MAPK [260]. Class II PI3K is also required for uterine smooth muscle contraction and pup delivery in mice [261]. 

## 10. Mechanical Control of PI3K Activity

The previous section was predicated on the assumption that PI3K lies upstream of RhoA/ROCK/myosin II signaling. However, recent evidence points to a possible regulatory circuit by which mechanical signals control PI3K signaling. In neuroepithelial cells, Eph depletion activates PI3K-AKT1 in a RhoA-dependent manner, inhibiting proliferation [262]. Likewise, blebbistatin (a myosin II ATPase inhibitor) altered PI3K activation in brain cells in response to TLR4 [263]. Interestingly, a recent study indicated that the two pathways are connected to control cellular stiffness [264], which is not only a marker of myosin II activation but also a hallmark of mechanical regulation, for example, for cancer emergence [265,266]. In this context, myosin IIA and MLCK are required for PI3K-AKT activation in MEK-inhibited triple-negative breast cells [267]. RhoA upstream of myosin II can also affect PI3K in other contexts, e.g., in human airway smooth muscle downstream of Gα12 [268]. Surprisingly, an ATPase-less isoform of myosin, myosin 18B, is required for PI3K activation in hepatocellular carcinoma [269], suggesting that myosin II filamentation, which depends on myosin 18B [270], may be important to transduce mechanical signals that mediate PI3K activation. Retrograde regulation seems to affect a wide array of processes beyond contraction, e.g., tissue factor expression is inhibited by blebbistatin and restored by wortmannin, indicating that myosin II lies upstream of PI3K to inhibit TF expression [271]. However, the highly heterogeneous nature of these reports strongly suggest that we have barely scratched the surface of the mechanical control of PI3K activity, which positions myosin II activation upstream of this signaling. While the current data does not support formulating an integrative model, an attractive possibility is that PI3K lies upstream of myosin II in short-term contraction (e.g., in response to chemoattractants) and physiological scenarios under continuous stress (e.g., smooth muscle contraction to control blood pressure), whereas mechanical control of PI3K activity, i.e., myosin II upstream of PI3K, may be important in chronic deformation of tissue, e.g., cancer-induced tissue stiffening.

## 11. Concluding Remarks

More than two decades after the discovery of the existence of crosstalk between Ras and Rho GTPases during oncogenic transformation and migration, the intricacies of their signaling network remain an exciting and active research area with many important open questions and implications in different diseases. Numerous targets can interact with Ras and Rho GTPases and extensive crosstalk and cooperation exist. The mechanisms underlying Ras and Rho GTPases crosstalk are not fully understood yet. Profound differences depending on the Ras isoform, whether the cells are epithelial or mesenchymal, have been reported, complicating an already convoluted picture. Adding to the complexity, it appears that certain signals depend on the kinetics and duration of Ras activation. The concerted and balanced action of many signaling molecules in both pathways make up for a highly complex signaling network with the ability to fine-tune the final cellular outcome depending on several parameters. Given the importance of Ras and Rho in many biological processes, the dynamics of their activation and termination, i.e., how their regulators and effectors are activated at the right time and place, and how this signal propagates in a variety of physiological and non-physiological processes, is an important unresolved issue. The generation and interrogation of mouse models that express oncogenic variants of Ras/Rho GTPases in certain tissues will be crucial to understand the interconnection of these pathways and will confirm or refute conflicting results observed in different cellular contexts. Given the relevance of both pathways in transformation and tumor progression, generation of these new models may lead to the identification of novel therapeutic targets within these signaling networks. Furthermore, increasing our understanding of the functional crosstalk between Ras and Rho signaling in the various tumor compartments will determine whether inhibition of this complex circuit may serve as effective treatment for newly diagnosed or recurrent tumors and will help identify optimum combinations of radiotherapy, cytotoxic chemotherapy, immunotherapy and other targeted molecular compounds.

## Figures and Tables

**Figure 1 genes-12-00819-f001:**
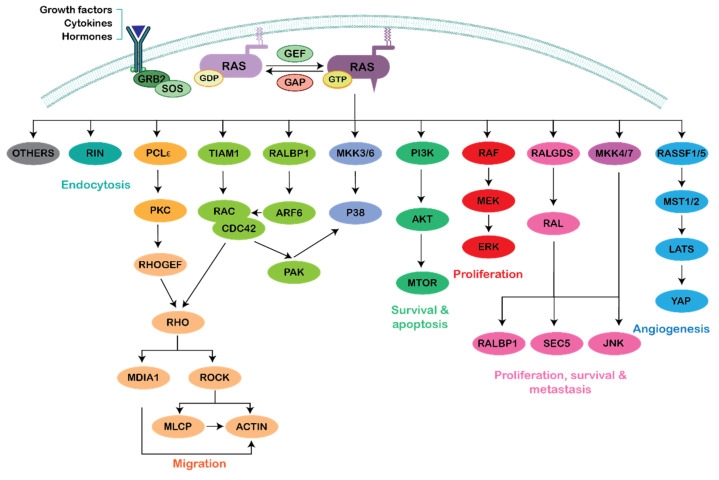
RAS effector pathways. Plasma membrane-associated Ras-GTP can directly interact with multiple, different effectors to activate various signaling cascades, regulating different cellular functions such as cell proliferation, migration, survival/death, differentiation, endocytosis migration and adhesion.

**Figure 2 genes-12-00819-f002:**
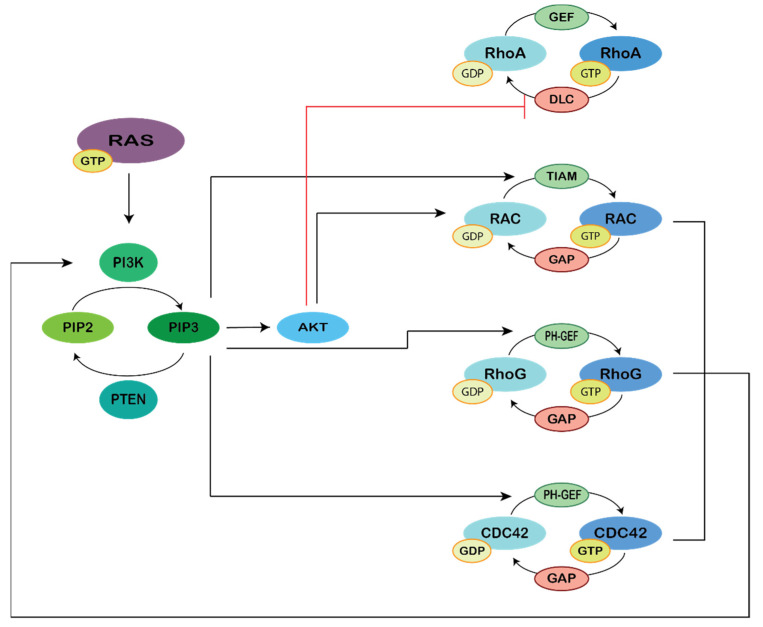
PI3K regulation of Rho-dependent signaling. Figure representing the complexity of Ras regulation of Rho by PI3K. Ras-GTP activates PI3K/AKT that, in turn, activates Rac to regulate actin polymerization and membrane ruffling. Active AKT inhibits DLC1 (a GAP for RhoA), increasing levels of RhoA, facilitating formation of focal adhesions. Formation of PtdIns(3,4,5)P3 (PIP3) recruits PH-containing Rho GEFs, including Tiam1, to the plasma membrane promoting activation of several Rho-GTPase members. PTEN and SHIP (not represented) dephosphorylates PIP3, thus antagonizing RAC activation by PIP3. Furthermore, combined activation of Rac, Cdc42 and RhoG potentiates activation of PI3K, creating a positive feedback on PI3K activation.

**Figure 3 genes-12-00819-f003:**
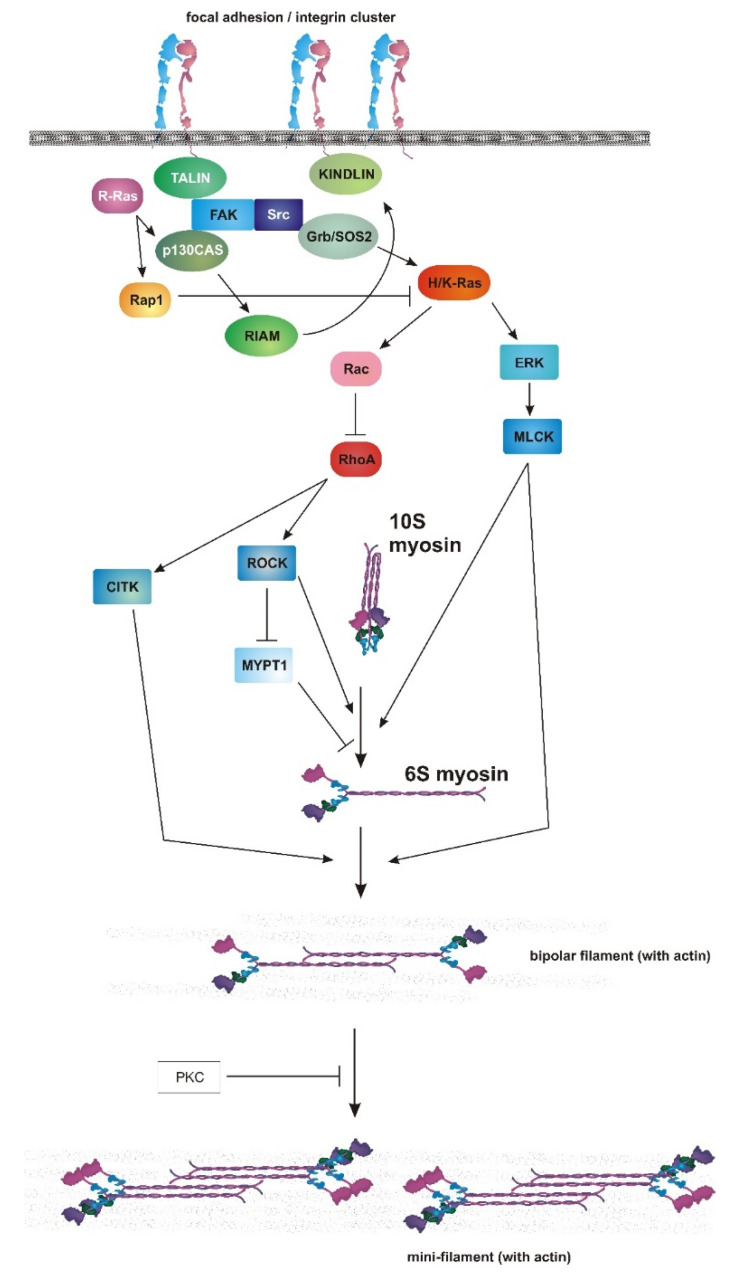
Control of myosin II activation by upstream signaling. Figure depicts diverse signaling pathways coming from and to the focal adhesions, including various Ras isoforms, and possible routes by which Ras, Rac and RhoA drive the different steps of myosin II activation, including conformational deployment (10S to 6S); the formation of the unstable species 6S, which forms bipolar filaments immediately; and the role of PKC in mini-filament stability. GTPases are shown in red-hued, round-edged boxes; adaptors in green-hued ellipses; and catalytic proteins in blue-hued boxes.

**Figure 4 genes-12-00819-f004:**
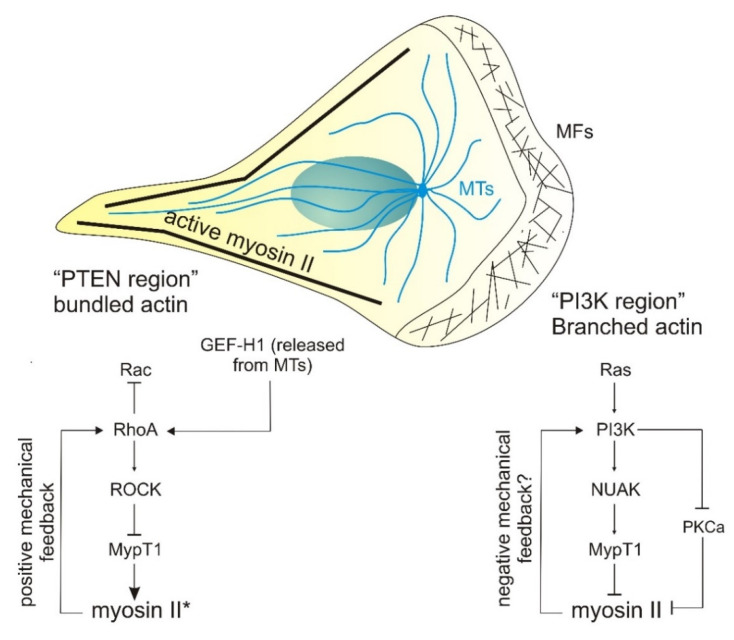
PI3K- and myosin II-based front–rear polarity in migrating cells. Diagram shows a polarized cell, migrating towards the right of the page. The protrusive (PI3K-enriched) region is devoid of filamentous myosin II and contains dendritic actin. Several mechanisms, including PI3K-dependent myosin inactivation by Nuak, enable protrusion. The left, non-protrusive pole accumulates the functional opposite of PI3K, PTEN, and bundled actomyosin under the control of RhoA. The possibility that a myosin II-dependent force controls the biochemical activation of these pathways is also contemplated as feedback loops.

## Data Availability

Not applicable.

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
