# Peer review of "The Crossroads between RAS and RHO Signaling Pathways in Cellular Transformation, Motility and Contraction"

_genes, 2021, doi:10.3390/genes12060819_

Round 1

Reviewer 1 Report

Soriano et al. is a comprehensive review on the intricacies of Ras and Rho signaling as drivers of oncogenic transformation. While this work is thorough and well researched, minor changes are necessary and should be addressed prior to publication. Recommendations are as follows:

Major Comments:

            Unfortunately, the appearance of the figures is very blurry/hard to read. Higher quality/resolution is needed. Figure titles/numbers should be included with the image itself.

Minor Comments:

            1. Citations needed on lines 40, 124, 70, 178, 250-255, 278-281, 349-356, 477-483, 501-504, 708-710

          2. Although it is discussed later in the manuscript, MLCK should be mentioned in addition to ROCK for phosphorylation of Myosin light chain on line 118.

           3.  Symbol error on lines 155, 218, 219, 220, 221, 333, 725

            4. Capitalization for Sos should remain consistent, used as SOS on line 390 and Sos on line 392 and throughout subsequent paragraph, SOS versus Sos1 should be clarified in paragraph beginning line 392. hSos needs to be defined prior to its first use and distinguished from ‘Sos’ and ‘Sos1’ used in this section.

           5.  Acronyms need to be defined throughout including: NSCLC (line 50), EGFR (line 53), RBD (line 154), MEFs (line 164), FGF2 (line 165), HGF (line 166), PDGF (line 176), WRC (line 193), PTEN (line 203), SHIP1 (line 203), WAVE2 (line 275), RACK1 (line 283), CRD (line 309), MRCK (line 642), RLC (line 644), DAPK (line 645)

          6.   Use of “)” is unclear in line 442

          7.   “in vivo” and “in vitro” should be italicized at each instance, ex. line 453

          8.   If p120RasGAP etc are going to be abbreviated as “p120” this needs to be defined at the first use and used consistently thereafter (lines 467-482). p120RasGAP is also written as “p120Ras-GAP” and “p120RASGAP”, consistency needed

Author Response

We thank reviewer 1 for his/her nice comments to our review and for the detailed revision. We have (hopefully) addressed all his/her concerns in this new version of the review.

Major Comments:

            Unfortunately, the appearance of the figures is very blurry/hard to read. Higher quality/resolution is needed. Figure titles/numbers should be included with the image itself.

We apologize for the low resolution of the initial figures. Same figures with increased resolution have now been inserted in the manuscript.  We have added figures titles and description as well.

Minor Comments:

  1. Citations needed on lines 40, 124, 70, 178, 250-255, 278-281, 349-356, 477-483, 501-504, 708-710

We thank the reviewer for this suggestion. Citations have been added to the mentioned parts.

  1. Although it is discussed later in the manuscript, MLCK should be mentioned in addition to ROCK for phosphorylation of Myosin light chain on line 118.

We have now added to the text his suggestion and final text now is: “ROCK, together with MLCK, acts by increasing the phosphorylation of myosin light chain, which…”

  1. Symbol error on lines 155, 218, 219, 220, 221, 333, 725

We apologise for this mistake on the original submission. All symbols have been revised and corrected.

  1. Capitalization for Sos should remain consistent, used as SOS on line 390 and Sos on line 392 and throughout subsequent paragraph, SOS versus Sos1 should be clarified in paragraph beginning line 392. hSos needs to be defined prior to its first use and distinguished from ‘Sos’ and ‘Sos1’ used in this section.

We apologise for inconsistency in the writing of Sos. We have now unified capitalization for Sos during the text. We have also clarified Sos vs Sos1 use by adding “Son of Sevenless (Sos) is a dual specificity GEF that regulates both Ras and Rho family GTPases and thus is uniquely poised to integrate signals that affect both gene expression and cytoskeletal reorganization. Two isoforms, Sos1 and Sos2 have been found, present-ing similar protein structures and cellular expression patterns, but specific functionalities (recently reviewed in Baltanas et al [132]).”

We apologise also for the misleading use of hSos in the text. It has now been substituted by Sos.

  1. Acronyms need to be defined throughout including: NSCLC (line 50), EGFR (line 53), RBD (line 154), MEFs (line 164), FGF2 (line 165), HGF (line 166), PDGF (line 176), WRC (line 193), PTEN (line 203), SHIP1 (line 203), WAVE2 (line 275), RACK1 (line 283), CRD (line 309), MRCK (line 642), RLC (line 644), DAPK (line 645)

We thank the reviewer for this comment. All acronyms have now been defined and we have also added an abbreviation section after the abstract that hope will facilitate reading of the review.

  1.  Use of “)” is unclear in line 442

We apologise for the misleading use of “)” in line 442. It has now been removed.

  1. “in vivo” and “in vitro” should be italicized at each instance, ex. line 453

We apologise for this mistake. “in vivo” and “in vitro” have been italicized now in all the text.

  1.  If p120RasGAP etc are going to be abbreviated as “p120” this needs to be defined at the first use and used consistently thereafter (lines 467-482). p120RasGAP is also written as “p120Ras-GAP” and “p120RASGAP”, consistency needed

We thank reviewer for this observation. p120RasGAP has been defined as p120 at the first use and p120 has been used thereafter.

Reviewer 2 Report

Soriano et al have written an excellent review on ""The Crossroads between Ras and Rho signaling pathways in cellular transformation, motility and contraction'". It is a well written review however there is a minor addition that the authors could include in their description which  would also mention the complexity of the field with mention on Ras and Rho independent signaling being active in cells and what it entails. Especially wrt to Ras independent signaling from the Barbacid lab.

Author Response

Soriano et al have written an excellent review on ""The Crossroads between Ras and Rho signaling pathways in cellular transformation, motility and contraction'". It is a well written review however there is a minor addition that the authors could include in their description which would also mention the complexity of the field with mention on Ras and Rho independent signaling being active in cells and what it entails. Especially wrt to Ras independent signaling from the Barbacid lab.

We would like to thank reviewer for his/her appreciation of our review and suggestion. We have now mentioned in the text the results obtained by Barbacid’s lab using Rasless fibroblasts, which clearly illustrate the interconnection between Ras and Rho function. More specifically, we have added a sentence in the introduction, in line 161 saying: “the activation of Rho GTPases leads to concurrent alterations in cell adhesion and cell motility [45] and loss of Ras proteins severely restricts cell motility and migration in fibro-blasts and cause major alterations in cytoskeletal structures [46].”

We have also added a paragraph in line 319: “Interestingly, Barbacid group found that in MEFs lacking Ras proteins (Rasless) only con-stitutive activation of components of the Raf/Mek/Erk pathway was sufficient to sustain normal migration. In a Rasless context, activation of the phosphatidylinositol 3-kinase (PI3K)/PTEN/Akt and Ral guanine exchange factor (RalGEF)/Ral pathways, either alone or in combination, failed to induce migration, although they cooperated with Raf/Mek/Erk signaling to reproduce the full response mediated by Ras signaling [46].”

Reviewer 3 Report

Authors wrote a very interesting and informative paper on describing the connection between RAS and Rho through PI3K activation, Raf/MEK/ERK activation, Ral GTPases. Authors describes various activation pathway via figures is quite impressive. However, I have some concerns which I have mentioned below.

  1. Authors should mention figure legends under every figure and cite it in text.
  2. In introduction, para 2, line 40, Please mention the reference “Aberrant Ras signalling is found in up to 30% of all human cancers….”
  3. There are some grammatical errors, authors need to review paper thoroughly.

Check for spelling “signaling” throughout the text.

Line 715, “RTK signalling in a glioma cell model in Drosophila led to the activation of Drak, a close…..”

Line 679, “Early evidence confirmed that the signalling routes activated…..”   

Author Response

We would like to thank the reviewer for his/her kind words to our review and appreciation of the figures. We have incorporated all the changes suggested in this new version of our review. We sincerely hope we have addressed all the concerns raised.

Authors wrote a very interesting and informative paper on describing the connection between RAS and Rho through PI3K activation, Raf/MEK/ERK activation, Ral GTPases. Authors describes various activation pathway via figures is quite impressive. However, I have some concerns which I have mentioned below.

1. Authors should mention figure legends under every figure and cite it in text.

We apologize for this mistake. Figure legends have now been added under every figure and cited in the text.

2. In introduction, para 2, line 40, Please mention the reference “Aberrant Ras signalling is found in up to 30% of all human cancers….”

We thank reviewer for this suggestion. A reference has now been added as suggested.

3. There are some grammatical errors, authors need to review paper thoroughly.

We have review the paper and hope the text does not have grammatical errors in this new version.

Check for spelling “signaling” throughout the text.

Line 715, “RTK signalling in a glioma cell model in Drosophila led to the activation of Drak, a close…”

Line 679, “Early evidence confirmed that the signalling routes activated…”   

We apologize for this mistake. “Signaling” has now been written throughout the text.

Round 2

Reviewer 3 Report

The authors have addressed all concerns, manuscript is good for publication.